# Electronic Beam Steering Metamaterial Antenna with Dual-Tuned Mode of Liquid Crystal Material

**DOI:** 10.3390/s23052556

**Published:** 2023-02-25

**Authors:** Shuang Ma, Xue-Nan Li, Zhan-Dong Li, Jun-Jun Ding

**Affiliations:** 1Department of Electronic Information Engineering, Shenyang Aerospace University, Shenyang 110034, China; 2Department of Civil Aviation, Shenyang Aerospace University, Shenyang 110034, China

**Keywords:** liquid crystal, metamaterial antenna, dual-tuned, beam steering

## Abstract

In this study, a dual-tuned mode of liquid crystal (LC) material was proposed and adopted on reconfigurable metamaterial antennas to expand the fixed-frequency beam-steering range. The novel dual-tuned mode of the LC is composed of double LC layers combined with composite right/left-handed (CRLH) transmission line theory. Through a multi-separated metal layer, the double LC layers can be loaded with controllable bias voltage independently. Therefore, the LC material exhibits four extreme states, among which the permittivity of LC can be varied linearly. On the strength of the dual-tuned mode of LC, a CRLH unit cell is elaborately designed on three-layer substrates with balanced dispersion values under arbitrary LC state. Then five CRLH unit cells are cascaded to form an electronically controlled beam-steering CRLH metamaterial antenna on a downlink Ku satellite communication band with dual-tuned characteristics. The simulated results demonstrate that the metamaterial antenna features’ continuous electronic beam-steering capacity from broadside to −35° at 14.4 GHz. Furthermore, the beam-steering properties are implemented in a broad frequency band from 13.8 GHz to 17 GHz, with good impedance matching. The proposed dual-tuned mode can make the regulation of LC material more flexible and enlarge the beam-steering range simultaneously.

## 1. Introduction

Over the past decade, liquid crystal (LC) has been introduced into the microwave and millimeter wave domain from optical and has experienced mushroom growth [1,2,3]. As a fire-new type of tunable anisotropic material, it attracts extensive attention over the superiority of its small size, light weight, low cost, wide application frequency range and easy conformity [4,5,6]. Up to now, LC has been comprehensively employed on tunable phase shifters [7], filters [8], resonators [9], antennas [10] and other microwave devices [11,12,13].

The composite right/left-handed (CRLH) transmission line is an artificial material which features unique electromagnetic properties that are unavailable in nature. Metamaterial antennas with an incorporated CRLH transmission line are capable of continuous beam steering from backfire to endfire, including broadside [14,15,16]. To date, CRLH metamaterial antennas have been extensively applied in smart antenna, satellite communication and high-resolution radar to promote the system performance [17,18,19]. Moreover, electronic beam-steering CRLH metamaterial antennas integrated tunable components or materials have the ability to beam steer in a fixed frequency and increase the frequency reuse ratio and the spectrum utilization [20,21,22]. Familiar modulation methods, including varactor [23], ferrite [24], BST [25], MEMS [26] and mechanical tuning, are broadly applied in microwave and millimeter wave components. Compared with the forenamed tuning methods, LC features a wide application frequency range, convenient feeding and continuous linear tuning [27,28]. However, tunable microwave devices based on LC suffer from the singular tunable mode and the restricted tunable scope [29,30]. There has been little agreement on how to fix these challenges [31,32,33]. Leaky wave antenna is a typical kind of beam-steering antenna. Due to its advantages of low profile, low cost and miniaturization, it has also become of great interest in recent years [34,35]. However, the traditional leaky wave antenna finds it difficult to break through the Bragg resonance effect; in consequence, it has trouble achieving broadside radiation. In addition, beam-steering antennas are good candidates for satellite communication to increase the channel capacity significantly and improve communication quality due to the beam-steering characteristics. Therefore, this paper focuses on the Ku band beam scanning antenna in the downlink band of satellite communication, combined with the dual-tuned scheme of LC material, to provide more possibilities for the application of the metamaterial antenna in satellite communication.

A novel dual-tuned mode of LC material is raised up and implemented on reconfigurable CRLH metamaterial antenna to achieve more flexible steering of the beam. In Section 2, the dual-tuned mode is illustrated and explained in detail. Section 3 describes the design, simulation and analysis of the dual-tuned beam-steering CRLH metamaterial antenna based on LC. Finally, the conclusion is summarized.

## 2. Dual-Tuned Mode of Liquid Crystal Material

The schematic diagram of an LC dual-tuned mode structure is shown in Figure 1. It consists of two parts: the main structure part and the support structure. The main structure part is an LC dual-tuned structure, which includes three substrate layers (Substrates A, B and C) and double LC layers (LC1 and LC2). A metal layer is deposited on the upper and lower surface of the middle substrate, Substrate B. The lower surface of Substrate A and the upper surface of Substrate C are coated with a metal layer. Two LC layers are sandwiched between the middle substrate, Substrate B, and the upper and the lower substrates, Substrates A and C. Substrates A and C are provided with small holes for filling LC material, and these holes can be sealed with glue after filling. Each metal layer on the dielectric substrate can be loaded with bias voltage separately. Furthermore, in order to fix the substrates, the non-dielectric support and the base are adopted as fixing devices on both of the laterals and the bottom. Small notches are machined on the inner sides of the support part to fix the substrates and ensure the thickness of the LC layers.

Nematic LCs are anisotropic materials composed of rod-like molecules, and they usually presents two permittivities along different directions: the parallel dielectric constant, *ε_r_*_//_ = 3.22 (in the direction of the LC molecules’ axes); the vertical dielectric constant, *ε_r_*_⊥_ = 2.43 (in the directions orthogonal to the LC molecules’ axes); and the loss tangent, *tanδ* = 0.006, which is derived from the material TUD-649 produced by Merck Corporation [29]. When LC material is in its natural state, the LC molecules are in a confused state, and the permittivity of LC is *ε_r_* = 2.83, as shown in Figure 1.

In the practical work of the LC, the dielectric constant along the *z*-axis direction mainly affects the dual-tuned mechanism; thus, the dielectric constant of the anisotropic LC material is expressed by the dielectric constant along the *z*-axis direction. When the LC layer is aligned along the *x*-axis (*V_DC_* = 0 V), LC molecules are parallel to the *x*-axis without the bias voltage, and the permittivity of the LC is *ε_r_*_⊥_ = 2.43, as depicted in Figure 2a. Upon the condition that the LC layer is loaded with bias voltage along the *z*-axis, the LC molecules tend to deflect from the direction of *x*-axis to *z*-axis as the bias voltage increases. At present, the dielectric constant is between the vertical dielectric constant and the parallel dielectric constant, *ε_r_*_⊥_ < *ε_r_* < *ε_r_*_∥_. When the bias voltage reaches the saturation value of the LC molecules (*V_DC_* = *V_max_*), the LC molecules are perpendicular to the direction of the *x*-axis and parallel to the direction of the *z*-axis, as sketched in Figure 2d. In this case, the permittivity of the LC is *ε_r_*_∥_ = 3.22. Upon the condition that the bias voltage is loaded on two LC layers, each LC layer presents two different states, as illustrated in Figure 2b,c. Thus, the dual-tuned mode makes the LC layer form four completely different extreme states. Meanwhile, it achieves continuing linear tuning between the four extreme states by modulating the magnitude of the bias voltage. Therefore, when the dual-tuned mode of the LC is adopted on metamaterial antennas, it provides a more diverse, more flexible beam-steering pattern and a wider beam-steering range of the metamaterial antenna.

## 3. Design, Simulation and Analysis of Dual-Tuned CRLH Metamaterial Antenna Based on LC

### 3.1. Design of CRLH Unit Cell

On the basis of the dual-tuned mode of LC material, a CRLH unit cell is elaborately designed, whose schematic diagram is sketched in Figure 3. Figure 3a shows the cross-section of the CRLH unit cell. It is composed of three substrates, four metal layers and two LC layers. From bottom to top, the first metal layer is the ground, while the second metal layer is the parallel inductor, as shown in Figure 3b, and the first LC layer is between the two metal layers. The third metal layer is an interdigital structure, as shown in Figure 3c, and the fourth metal layer is the cover sheet; the second LC layer is between them. The second metal layer is connected to the third metal layer by a metallized via hole to formulate left-handed parallel inductance. The series capacitor is composed of two groups of series interdigital structures. In order to facilitate the feeding, the insertion of the finger is connected with a strip line. The corresponding equivalent circuit diagram of the CRLH unit cell is demonstrated in Figure 3d. The detailed parameters of the structure and the equivalent circuit (LC under natural state) in Figure 3 are listed in Table 1.

### 3.2. Theoretical Analysis of CRLH Structure with Dual-Tuned Mode

According to the antenna theory [36], the beam direction of the antenna *θ* is determined by the phase constant, *β*, and wave number in free space, *k*_0_, as follows:(1)$$\theta =\arccos\frac{\beta }{k_{0}}$$

In the CRLH metamaterial antenna, the phase constant, *β*, can be expressed as follows [37]:(2)$$\beta =\omega \sqrt{L_{R}C_{R}}-\frac{1}{\omega \sqrt{L_{L}C_{L}}}$$
where *ω* is the angular frequency of the antenna, *L_R_* and *C_R_* are the right-handed inductance and capacitance, and *L_L_* and *C_L_* are the left-handed inductance and capacitance. When the LC is incorporated into the electronic beam-steering metamaterial antenna, the *L_R_*, *C_R_*, *L_L_* and *C_L_* can be transformed with the tuning of the LC state.

When the LC material is under the alignment state and the saturation bias voltage, sequentially, the dispersion functions of the CRLH unit cell are described as follows:(3)$$\beta_{0}\left(\omega \right)=\omega \sqrt{L_{R0}C_{R0}}-\frac{1}{\omega \sqrt{L_{L0}C_{L0}}}$$
(4)$$\beta_{V}\left(\omega \right)=\omega \sqrt{L_{RV}C_{RV}}-\frac{1}{\omega \sqrt{L_{LV}C_{LV}}}$$
where *β*_0_ is the dispersion of the CRLH unit cell; *L_R_*_0_ and *C_R_*_0_ are the right-handed inductance and capacitance of the CRLH unit cell; *L_L_*_0_ and *C_L_*_0_ are the left-handed inductance and capacitance of the CRLH unit cell when LC is under the alignment state; *β_V_* is the dispersion of the CRLH unit cell; *L_RV_* and *C_RV_* are the right-handed inductance and capacitance of the CRLH unit cell; and *L_LV_* and *C_LV_* are the left-handed inductance and capacitance of the CRLH unit cell when the LC is loaded with saturated bias voltage.

Then the beam-scanning angle of the CRLH unit cell, Δ*θ*, can be expressed as follows:(5)$$\Delta \theta =\arcsin\left(\frac{\omega }{k_{0}}\sqrt{L_{RV}C_{RV}}-\frac{1}{\omega k_{0}\sqrt{L_{LV}C_{LV}}}\right)-\arcsin\left(\frac{\omega }{k_{0}}\sqrt{L_{R0}C_{R0}}-\frac{1}{\omega k_{0}\sqrt{L_{L0}C_{L0}}}\right)$$

Equation (5) is written as follows:(6)$$k_{0}\left(\sin\theta_{V}-\sin\theta_{0}\right)=\omega \left(\sqrt{L_{RV}C_{RV}}-\sqrt{L_{R0}C_{R0}}\right)+\frac{\sqrt{L_{LV}C_{LV}}-\sqrt{L_{L0}C_{L0}}}{\omega \sqrt{L_{LV}C_{LV}L_{L0}C_{L0}}}$$

According to the dual-tuned mode of the LC in Section 2, when the bias voltage is loaded onto different LC layers, the dual-tuned mode shows four different states, thus causing the metamaterial antenna based on the LC to form beam steering. The phase constants corresponding to the four extreme states of the dual-tuned mode can be expressed as follows:(7)$$\beta_{00}\left(\omega \right)=\omega \sqrt{L_{R00}C_{R00}}-\frac{1}{\omega \sqrt{L_{L00}C_{L00}}}$$
(8)$$\beta_{V0}\left(\omega \right)=\omega \sqrt{L_{RV0}C_{RV0}}-\frac{1}{\omega \sqrt{L_{LV0}C_{LV0}}}$$
(9)$$\beta_{0V}\left(\omega \right)=\omega \sqrt{L_{R0V}C_{R0V}}-\frac{1}{\omega \sqrt{L_{L0V}C_{L0V}}}$$
(10)$$\beta_{VV}\left(\omega \right)=\omega \sqrt{L_{RVV}C_{RVV}}-\frac{1}{\omega \sqrt{L_{RVV}C_{RVV}}}$$
where *β*_00_, *L_R_*_00_, *C_R_*_00_, *L_L_*_00_ and *C_L_*_00_ are the phase constant, the right-handed inductance, the right-handed capacitance, the left-handed inductance and the left-handed capacitance of the CRLH unit cell when two LC layers are both under the alignment state; *β*_0*V*_, *L_R_*_0*V*_, *C_R_*_0*V*_, *L_L_*_0*V*_ and *C_L_*_0*V*_ are the phase constant, the right-handed inductance, the right-handed capacitance, the left-handed inductance and the left-handed capacitance of the CRLH unit cell when LC1 layer is under the alignment state and the LC2 layer is loaded with saturated bias voltage; *β_V_*_0_, *L_RV_*_0_, *C_RV_*_0_, *L_LV_*_0_ and *C_LV_*_0_ are the phase constant, the right-handed inductance, the right-handed capacitance, the left-handed inductance and the left-handed capacitance of the CRLH unit cell when the LC1 layer is loaded with saturated bias voltage and the LC2 layer is under the alignment state; and *β_VV_*, *L_RVV_*, *C_RVV_*, *L_LVV_* and *C_LVV_* are the phase constant, the right-handed inductance, the right-handed capacitance, the left-handed inductance and the left-handed capacitance of the CRLH unit cell when the two LC layers are both loaded with saturated bias voltage.

Then, under the two extreme states (two LC layers under alignment state and loaded with saturated bias voltage), Equation (6) can be expressed as follows:(11)$$\begin{array}{c} k_{0}\left(\sin\theta_{VV}-\sin\theta_{00}\right)=\omega \left(\sqrt{L_{RVV}(C_{R1VV}+C_{R2VV})}-\sqrt{L_{R00}(C_{R100}+C_{R200})}\right)\\ +\frac{\sqrt{L_{LVV}C_{LVV}}-\sqrt{L_{L00}C_{L00}}}{\omega \sqrt{L_{LVV}C_{LVV}L_{L00}C_{L00}}} \end{array}$$
where *θ*_00_ is the main lobe direction of antenna; *C**_R_*_100_ and *C**_R_*_200_ are the right-hand capacitance formed by LC1 and the right-handed capacitance formed by LC2 when both layers of LC are in alignment; *θ**_VV_* is the main lobe direction; and *C**_R_*_1*VV*_ and *C**_R_*_2*VV*_ are the right-handed capacitance formed by LC1 and the right-handed capacitance formed by LC2 when both layers of LC are in the state of saturated bias voltage.

Since the right-handed inductance is invariable with the tuning of the LC state, *L_R_* can be used to replace the right-handed inductance in any state, and the above formula is simplified to obtain the following formula:(12)$$\begin{array}{l} k_{0}\left(\sin\theta_{VV}-\sin\theta_{00}\right)=\\ \omega \sqrt{L_{R}}\left(\sqrt{C_{R1VV}+C_{R2VV}}-\sqrt{C_{R100}+C_{R200}}\right)+\frac{1-\sqrt{\frac{L_{L00}C_{L00}}{L_{LV}C_{LV}}}}{\omega \sqrt{L_{L00}C_{L00}}} \end{array}$$

Employing the same simplified method to process the expression of the single-tuned CRLH LC metamaterial antenna, Equation (12) can be rewritten as follows:(13)$$k_{0}\left(\sin\theta_{V}-\sin\theta_{0}\right)=\omega \sqrt{L_{R}}\left(\sqrt{C_{RV}}-\sqrt{C_{R0}}\right)+\frac{1-\sqrt{\frac{L_{L0}C_{L0}}{L_{LV}C_{LV}}}}{\omega \sqrt{L_{L0}C_{L0}}}$$

From Equations (12) and (13), for the dual-tuned LC CRLH metamaterial antenna, the maximum beam scanning range is Δ*θ*’ = *θ_VV_* − *θ*_00_, and the maximum beam scanning range of the single-tuned CRLH LC metamaterial antenna is Δ*θ* = *θ_V_* − *θ*_0_.

Compared with the beam scanning range expression of a single-tuned CRLH LC metamaterial antenna, in a dual-tuned LC metamaterial antenna, the change rate of the right-hand capacitance is enhanced due to the addition of the second LC layer. Thus, the phase constant of the right-handed transmission line improves accordingly, meaning that the first item on the right of the equal sign is enlarged. Therefore, the beam scanning range of the metamaterial antenna is improved. In addition, due to the addition of variable left-handed inductance in the dual-tuned LC metamaterial antenna, the change rate of the phase constant of the left-handed transmission line also increases. At the same time, the phase constant of the composite left-handed transmission line is expended, which is the second term on the right of the equal sign. Hence, the beam scanning range of the metamaterial antenna is amplified. To sum up, in theory, the dual-tuned CRLH structure can improve the change rate of the phase constant both of the left-handed and the right-handed transmission.

The dual-tuned CRLH unit cell model is simulated and analyzed by CST electromagnetic simulation software. DC bias voltages *V*_DC1_ and *V*_DC2_ are loaded on two LC layers, respectively, leading to four different extreme states of LC, as shown in Figure 2. The simulation results of the CRLH unit cell are shown in Figure 4 and Figure 5 under the four different states. When no bias voltage is loaded and the LC layers are aligned, the balanced frequency is 14.4 GHz, where the dispersion is zero. While the LC1 is loaded with the saturation bias voltage, and the LC2 is aligned, the left-handed capacitance is decreased, giving rise to the variation of the balanced frequency to 14.65 GHz. Then, if the LC1 is aligned and the LC2 is loaded with the saturation bias voltage, the right-handed capacitance is enlarged and the left-handed inductance is diminished, the balanced frequency is shifted to 14.78 GHz. When the two LC layers are loaded with the saturation bias voltage, both the left-handed capacitance and the left-handed inductance are reduced, while the right-handed capacitance is enhanced, corresponding to the movement of the balanced frequency from 14.78 GHz to 15.1 GHz. The dispersion curves of *βd* via frequency in Figure 4 are compared with numerical calculations of parameters in the equivalent circuit, which is represented by discrete points. The theoretical calculation and simulation results are in good agreement, thus demonstrating the correctness of the anterior theoretical analysis. Moreover, the results illustrate that the CRLH unit cell keeps balance states as the bias voltage changes all along.

### 3.3. Simulation of CRLH Metamaterial Antenna Based on LC

Afterward, five CRLH unit cells are cascaded to form an electronically controlled beam-steering CRLH metamaterial antenna based on the dual-tuned mode of LC, as displayed in Figure 6. In order to let the main structure of the antenna look more distinguished, the upper substrate (Substrate A), the metal cover sheet, substrate C and the ground metal layer in Figure 1 are omitted. The intermediate substrate (Substrate B) in Figure 1 is set to be semitransparent. The parallel inductor is below Substrate B, while the interdigital structure is above Substrate B. From the figure, the position relationship between the parallel inductor and the upper series interdigital capacitor can be seen more distinctly, and they are connected through the metallized via holes. Two ports are adopted to form a traveling wave antenna to avoid reflection and standing wave, which are on the left and right sides of the antenna.

For the dual-tuned LC structure, the bias voltage of the first LC layer is loaded between the cover sheet and the interdigital structure, and the bias voltage of the second LC layer is loaded between the interdigital structure (loaded to the parallel inductor through the metallized via) and the ground. When Port 1 is fed and Port 2 is connected to matching load, due to the dual-tuned mode of the LC, the electronically controlled beam-steering CRLH metamaterial antenna has beam-steering characteristics at different bias voltages, as displayed in Figure 7. From Figure 7a, at the balanced frequency 14.4 GHz, the simulated realized far-field pattern shifts from 0° to −35°, including −8° and −18°, with the side lobe below −10 dB. Meanwhile, the realized gain is maintained around 6 dB. Due to the continuous scanning characteristic of the LC, the CRLH metamaterial antenna can also achieve continuous beam steering from 0° to −35°. In the case that the feed is varied to Port 2 while Port 1 is connected to the matching load, a beam scanning range from 0 to 35 degrees can be achieved. From the curve of the main lobe direction via frequency in Figure 7b, with the redshift of the balanced frequency, the broadside frequency also transforms. In addition to acquiring fixed-frequency beam steering, CRLH metamaterial antenna also has frequency scanning characteristics from negative to positive, including broadside in the range of 2 GHz (from 14 GHz to 16 GHz) in each LC state. For the CRLH structure, the zero point of the phase of the unit cell corresponds to the frequency point of the antenna broadside. Thus, under the four extreme states of the LC, the frequency point of the zero phase in the unit cell dispersion simulation results is compared with the frequency point when the antenna beam is pointing at 0°, as shown in Table 2. It can be seen from the table that, in any LC state, the frequency point of phase zero is consistent with the corresponding broadside frequency of the antenna. There is no frequency offset, and the simulation results have good consistency. The radiation efficiency of the antenna under the four extreme states is displayed in Figure 7c. Due to the employment of multilayers and LCs, the radiation efficiency of the antenna is around 40% at 14.4 GHz. In the meantime, the magnitude of S_11_ keeps below −10 dB, which has good impedance, matching in a broad frequency band from 13.8 GHz to 17 GHz, as presented in Figure 8. Meanwhile, the S21 curves of the antenna are also described in Figure 8 to demonstrate the transmission of the electromagnetic wave. As shown in the figure, on account of the reflection of the incident port, the loss of the substrate and the LC material, and the radiation of the antenna, fewer electromagnetic waves are transmitted from Port 2. Moreover, the parameters of the proposed antenna are compared with other LC-based antennas in the references in Table 3. As shown in the table, the performance of the LC metamaterial antenna proposed in this paper is relatively balanced in all aspects. In addition, it also has the advantages of flexible and diversified tuning modes.

## 4. Conclusions

An electronic beam-steering CRLH metamaterial antenna loaded with nematic LC material on a downlink Ku satellite communication band was presented in this paper. The dual-tuned mode of the LC is proposed through the combination of three substrates with multi-layers of metal etching and double LC layers between them. The bias voltages are loaded on the multi-metal layers, sequentially, to control the electric field on the LC layers and deflect the direction of the LC molecules. Thus, the LC material exhibits four extreme states by appropriate bias voltages, and the permittivity transforms linearly between these states. A CRLH unit cell is designed based on the dual-tuned mode of the LC and CRLH transmission line theory. Moreover, an electronically controlled beam-steering metamaterial antenna is constituted by the CRLH unit cells. The simulation results reveal that the antenna can achieve a 35° beam steering range, including broadside at 14.4 GHz with 6 dB stable realized gain under the dual-tuned mode. In addition, the antenna realizes fixed-frequency beam steering at any frequency within the 2 GHz band of the Ku downlink band. This paper provided a fire-new approach for the tuning mode of LC and supplied a theoretical basis for the future development of LC microwave devices.

## Figures and Tables

**Figure 1 sensors-23-02556-f001:**
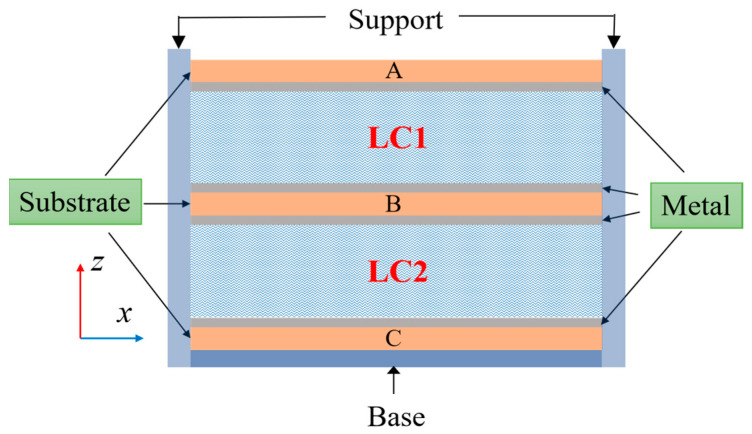
Schematic diagram of the LC dual-tuned mode structure.

**Figure 2 sensors-23-02556-f002:**
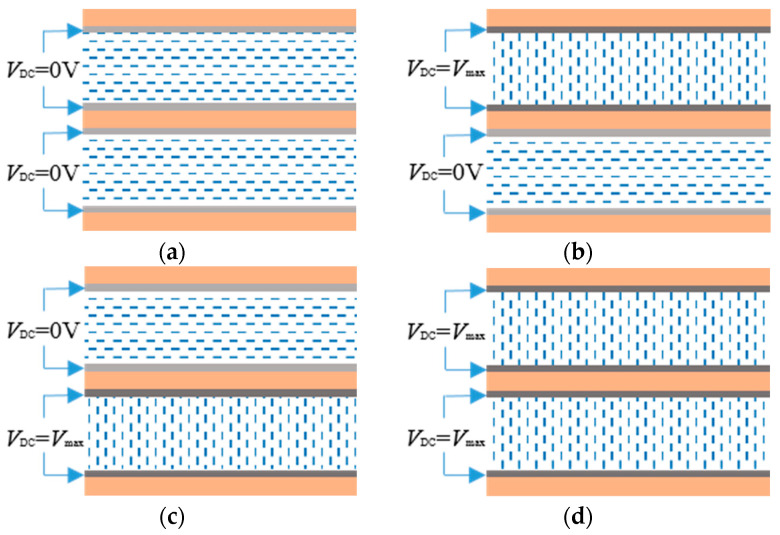
Schematic diagram of four different states of dual-tuned mode based on (**a**) alignment of two LC layers, (**b**) loading bias voltage on the first LC layer, (**c**) loading bias voltage on the second LC layer and (**d**) loading bias voltage on two LC layers.

**Figure 3 sensors-23-02556-f003:**
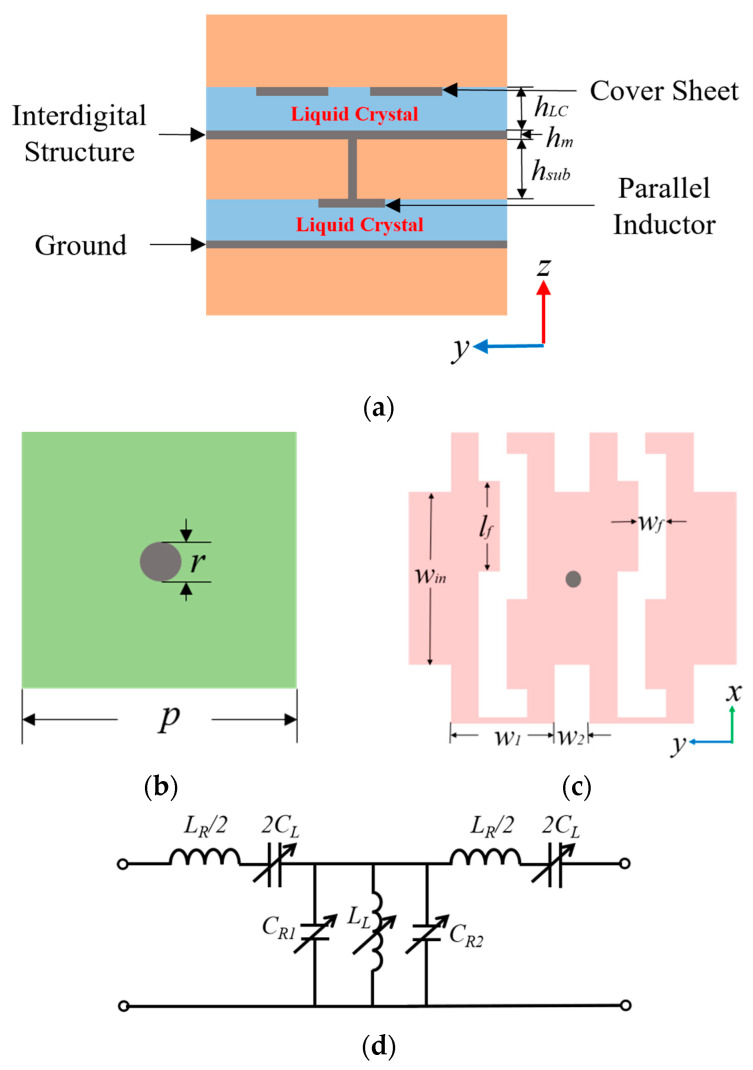
Schematic diagram of the CRLH unit cell: (**a**) cross-section, (**b**) the parallel inductor structure, (**c**) the interdigital structure and (**d**) equivalent circuit diagram of the CRLH unit cell.

**Figure 4 sensors-23-02556-f004:**
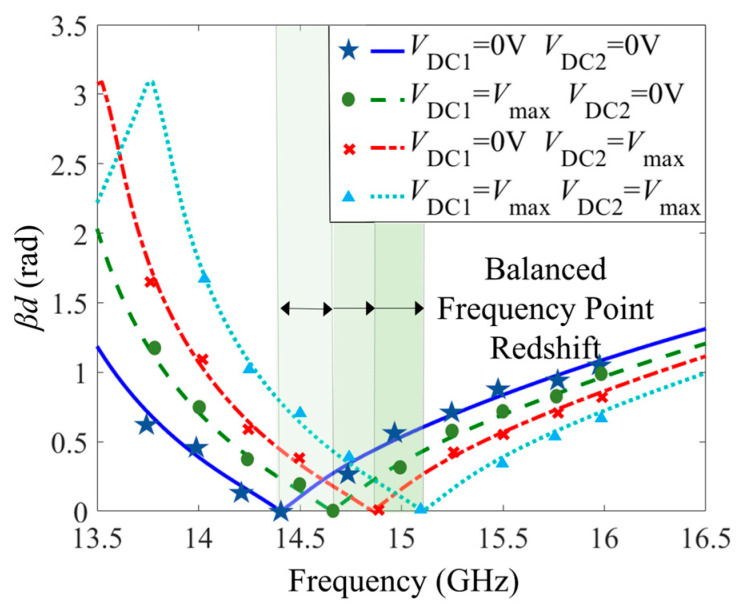
*βd* of the CRLH unit cell under dual-tuned mode of LC.

**Figure 5 sensors-23-02556-f005:**
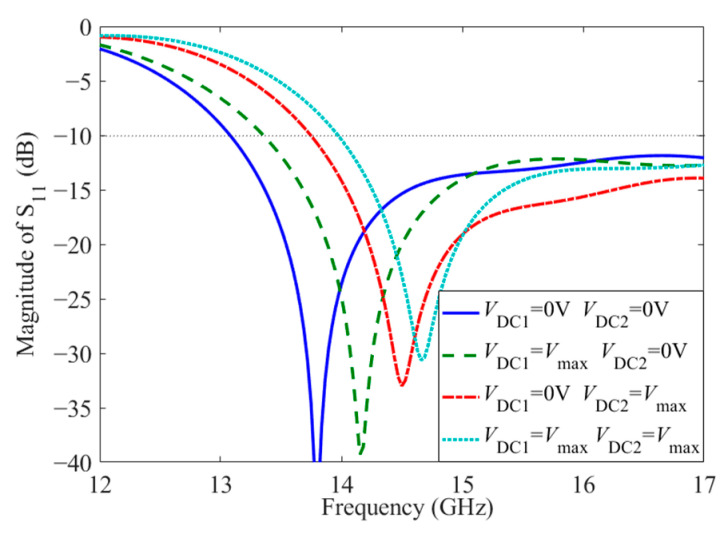
Magnitude of S_11_ of the CRLH unit cell under dual-tuned mode of LC.

**Figure 6 sensors-23-02556-f006:**
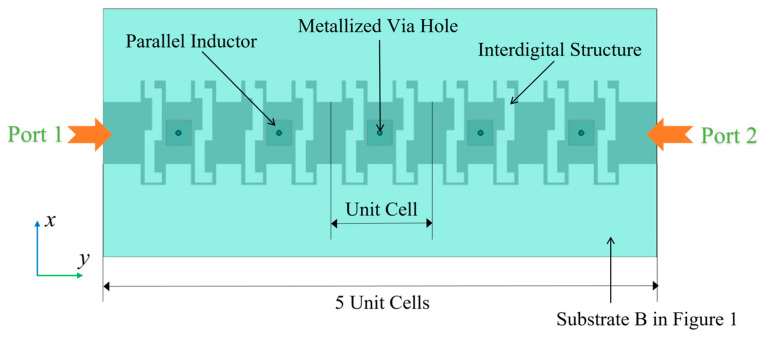
Schematic diagram of the CRLH metamaterial antenna based on LC.

**Figure 7 sensors-23-02556-f007:**
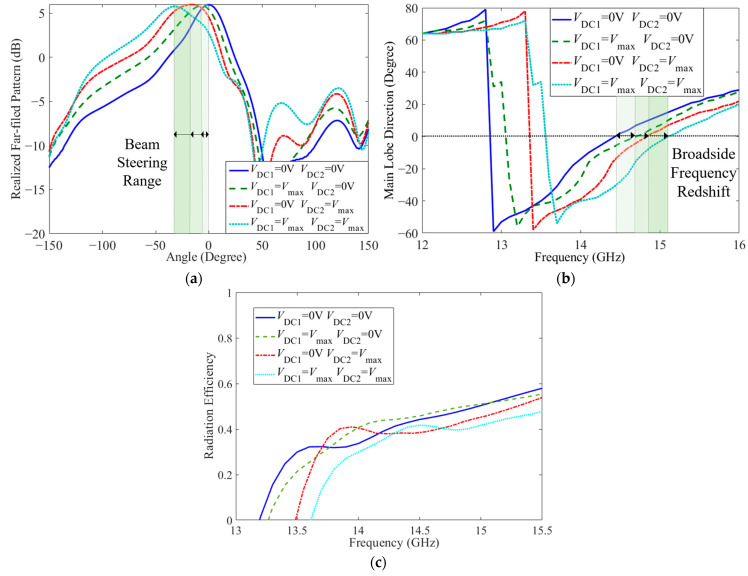
Radiation of the CRLH metamaterial antenna: (**a**) the realized far-field pattern at 14.4 GHz under dual-tuned mode of LC, (**b**) curve of the main lobe direction varies with frequency under dual-tuned mode of LC and (**c**) radiation efficiency of the antenna.

**Figure 8 sensors-23-02556-f008:**
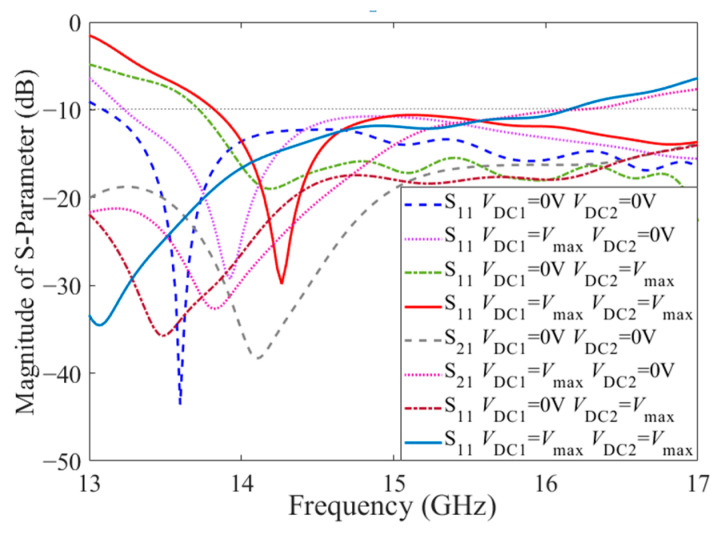
Magnitude of S_11_ and S_21_ of the electrical beam-steering CRLH metamaterial antenna under dual-tuned mode of LC.

**Table 1 sensors-23-02556-t001:** Parameters of the CRLH unit cells.

Parameter	*h_LC_*	*h_m_*	*h_sub_*	*p*	*r*	*L_R_*	*L_L_*
Value	0.25 mm	0.018 mm	0.8 mm	2.5 mm	0.2 mm	1.02 nH	0.5 nH
Parameter	*w_f_*	*w_1_*	*w_2_*	*l_f_*	*w_in_*	*C_R_*	*L_L_*
Value	0.4 mm	1.5 mm	0.5 mm	1.3 mm	2.5 mm	0.625 pF	0.04 pF

**Table 2 sensors-23-02556-t002:** Comparison for the zero frequency between the unit cell and the antenna.

Bias Voltage State	CRLH Unit CellFreq. (*βd* = 0) (GHz)	AntennaFreq. (*θ* = 0) (GHz)
*V*_DC1_ = 0 V, *V*_DC2_ = 0 V	14.4	14.42
*V*_DC1_ = *V*_max_, *V*_DC2_ = 0 V	14.65	14.66
*V*_DC1_ = 0 V, *V*_DC2_ = *V*_max_	14.78	14.8
*V*_DC1_ = *V*_max_, *V*_DC2_ = *V*_max_	15.1	15.11

**Table 3 sensors-23-02556-t003:** Comparison with other LC antennas.

Ref.	Type	Freq. (GHz)	Max S_11_ (dB)	Beam Range (°)	Gain (dB)
[38]	CRLH	27	−2	40	-
[39]	Rectangular Waveguide	9.7	−7	31	7
[40]	CRLH	26.7	−8	14	-
[41]	Half Mode SIW	15.8	−12.5	22	9.8
[42]	Half Mode SIW	11	−16	25	10
This Work	CRLH	14.4	−11	35	6

## Data Availability

The study did not report any data.

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
