# Peer review of "Electronic Beam Steering Metamaterial Antenna with Dual-Tuned Mode of Liquid Crystal Material"

_sensors, 2023, doi:10.3390/s23052556_

Round 1

Reviewer 1 Report

Dear Authors,

You have responded well to my previous observations. 

I have no further comments.

Author Response

Dear Reviewer,

Thank you for your approval and look forward to more exchanges in the future.

Best Regards,

Shuang Ma & Zhan-Dong Li

Reviewer 2 Report

The authors revised the manuscript according to my comments. Therefore, I accept the manuscript to be published on Sensors.

Author Response

(The authors gave the same response as above.)

Reviewer 3 Report

I think that the revised manuscript is well constructed with the response of the authors. It can be published in its current form.

Author Response

(The authors gave the same response as above.)

Reviewer 4 Report

Authors in this research article have presented and investigated Electronic Beam Steering Metamaterial Antenna with Dual-tuned Mode of Liquid Crystal Material. The topic and concept of the paper are interesting and it includes promising results. Prior to final acceptance recommendation the authors are encouraged to address the following comments.

1.      Its language needs some minor modifications.

2.      Determine the relationship between inductor and capacitor with changes in dielectric constant

3.      Considering that the structure is CRLH, in Figure 7, why the authors did not report the pattern with positive scan angle, Discuss this topic or show the results in the figure.

4.      The radiation efficiency of the antenna has not been checked. Please draw its diagram.

5.      Considering that the antenna is two-port, why is the transmission loss diagram not drawn?

6.      A comparison table with similar articles should be given

7.     The provided antenna is one of the adjustable leaky wave antennas with negative to positive scan angle. But the authors have not mentioned much about such antennas in the introduction. The following articles can be used to explain more about adjustable leakage wave antennas:

-"Microwave Liquid Crystal Enabling Technology for Electronically Steerable Antennas in SATCOM and 5G Millimeter-Wave Systems"

-"Electronic Beam-Scanning Antenna Based on a Reconfigurable Phase-Modulated Metasurface"

-“Magnetically Scannable Slotted Waveguide Antenna based on the Ferrite with Gain Enhancement”

-“Substrate integrated waveguide leaky wave antenna with circular polarization and improvement of the scan angle”

Author Response

Dear Reviewer,

        Thank you very much for your careful review of our manuscript. We addressed all questions, concerns, and suggestions in the following response. In addition, significant changes have been made to the manuscript and a new version has been submitted with this letter.

        Changed text passages are highlighted in blue, and the main revisions are listed as follow:

  • Expression of the manuscript is carefully checked and revised.
  • The relationship between the inductor and the capacitor when the dielectric constant changes is more fully described.
  • The pattern of the positive scanning angle is further illustrated.
  • The radiation efficiency of the antenna is displayed in Figure 7(c).
  • The transmission loss is supplemented in Figure 8 together with the return loss.
  • A comparison table of similar articles is added in Table 3 for better completeness.
  • The references were adjusted and improved in detail.

Details in the changes are mentioned in corresponding answers to reviewers’ comments in the attachment.

For further question please don’t hesitate to contact us.

Shuang Ma & Zhan-Dong Li

Shenyang Aerospace University, Shenyang, China

Round 2

Reviewer 4 Report

The authors have answered the referees well and this article can be published in this journal